# Deep learning analysis of epicardial adipose tissue to predict cardiovascular risk in heavy smokers
Borek Foldyna [1] ✉, Ibrahim Hadzic [2,3,4], Roman Zeleznik[2,3], Marcel C. Langenbach[1,5],
Vineet K. Raghu [1], Thomas Mayrhofer [1,6], Michael T. Lu [1] & Hugo J. W. L. Aerts [2,3,4]

## Abstract

**Background** Heavy smokers are at increased risk for cardiovascular disease and may benefit from individualized risk quantification using routine lung cancer screening chest computed tomography. We investigated the prognostic value of deep learning-based automated epicardial adipose tissue quantification and compared it to established cardiovascular risk factors and coronary artery calcium.

**Methods** We investigated the prognostic value of automated epicardial adipose tissue quantification in heavy smokers enrolled in the National Lung Screening Trial and followed for 12.3 (11.9–12.8) years. The epicardial adipose tissue was segmented and quantified on non-ECG-synchronized, non-contrast low-dose chest computed tomography scans using a validated deep-learning algorithm. Multivariable survival regression analyses were then utilized to determine the associations of epicardial adipose tissue volume and density with all-cause and cardiovascular mortality (myocardial infarction and stroke).

**Results** Here we show in 24,090 adult heavy smokers (59% men; 61 ± 5 years) that epicardial adipose tissue volume and density are independently associated with all-cause (adjusted hazard ratios: 1.10 and 1.38; $P < 0.001$) and cardiovascular mortality (adjusted hazard ratios: 1.14 and 1.78; $P < 0.001$) beyond demographics, clinical risk factors, body habitus, level of education, and coronary artery calcium score.

**Conclusions** Our findings suggest that automated assessment of epicardial adipose tissue from low-dose lung cancer screening images offers prognostic value in heavy smokers, with potential implications for cardiovascular risk stratification in this high-risk population.

## Plain Language Summary

Heavy smokers are at increased risk of poor health outcomes, particularly outcomes related to cardiovascular disease. We explore how fat surrounding the heart, known as epicardial adipose tissue, may be an indicator of the health of heavy smokers. We use an artificial intelligence system to measure the heart fat on chest scans of heavy smokers taken during a lung cancer screening trial and following their health for 12 years. We find that higher amounts and denser epicardial adipose tissue are linked to an increased risk of death from any cause, specifically from heart-related issues, even when considering other health factors. This suggests that measuring epicardial adipose tissue during lung cancer screenings could be a valuable tool for identifying heavy smokers at greater risk of heart problems and death, possibly helping to guide their medical management and improve their cardiovascular health.

Individuals eligible for lung cancer screening have a high risk of dying from cardiovascular (CV) causes, even higher than lung cancer[1–3]. While low-dose screening chest computed tomography (CT) is recommended to detect lung cancer, the heart is also imaged but not the focus of the evaluation. Thus, screening CTs offer a unique opportunity to assess imaging markers to predict CV disease (CVD) risk without additional radiation or cost.

Although prior studies have shown a relationship between lung screening CT-derived coronary artery calcium (CAC) and mortality[4], data

on the prognostic value of other imaging features is limited. Epicardial adipose tissue (EAT) is a metabolically active organ directly adjacent to coronary arteries[5,6] that has paracrine proatherogenic effects on coronary artery walls[7,8]. A growing body of evidence links increased EAT volume and density to coronary artery disease (CAD) and adverse CV events in asymptomatic[9–11], stable chest pain[12], and acute chest pain[13,14] individuals. Most prognostic data is based on manual, time-consuming EAT measurements using specialized electrocardiogram (ECG)-synchronized

[1]Cardiovascular Imaging Research Center (CIRC), Massachusetts General Hospital, Harvard Medical School, Boston, MA, USA. [2]Artificial Intelligence in Medicine (AIM) Program, Mass General Brigham, Harvard Medical School, Boston, MA, USA. [3]Department of Radiation Oncology, Brigham and Women's Hospital, Dana-Farber Cancer Institute, Harvard Medical School, Boston, MA, USA. [4]Radiology and Nuclear Medicine, CARIM & GROW, Maastricht University, Maastricht, The Netherlands. [5]Institute for Diagnostic and Interventional Radiology, University Hospital Cologne, Cologne, Germany. [6]School of Business Studies, Stralsund University of Applied Sciences, Stralsund, Germany. ✉e-mail: bfoldyna@mgh.harvard.edu

cardiac CT scans. The prognostic value of automated EAT assessment from non-gated low-dose chest CTs is unknown but of substantial clinical relevance, given the high number of individuals undergoing lung cancer screening and their increased CVD risk. Hence, an automated EAT assessment has the potential of becoming a new, low-cost, low-effort, and high-yield tool to identify individuals at increased CVD risk, guide their primary prevention, and improve their CV health.

The primary objective of this study was to investigate whether an automated deep-learning algorithm to measure EAT volume and density on non-ECG-gated low-dose lung screening chest CTs helps predicting all-cause and CV mortality in a high-risk population of heavy smokers. This study shows that automatically assessed EAT volume and density relate to all-cause and CV mortality, independent of traditional CV risk factors, including level of education, body habitus, and CAC score.

## Methods
### Study setting and participants
Our study retrospectively analyzed prospectively acquired data of 24,090 individuals enrolled in the CT arm of the National Lung Screening Trial (NLST; ClinicalTrials.gov identifier: NCT00047385), a community-based randomized controlled trial of chest X-rays vs. low-dose chest CT imaging for lung cancer screening[1,15]. The NLST trial included current/recent (smoking cessation within the past 15 years) and former heavy smokers with a 30-pack-year or more smoking history aged 55 to 74 years from 33 U.S. sites between August 2002 and April 2004[15]. The study size was driven by the available NLST data and included all participants randomized to the CT arm. Participants without available or interpretable CTs were excluded. Flow diagram Supplementary Fig. 1 provides specific study cohort details. All individuals were prospectively followed over 12 years for incident adverse events, including all-cause and CV mortality (fatal myocardial infarction or stroke)[1]. In addition, self-reported sex, race, and ethnicity were acquired as a part of the parent trial.

The secondary use of NLST data was approved by the National Cancer Institute (Bethesda, Maryland) and the local Mass General Brigham (Boston, MA) institutional review board (IRB#: 2017P002844). All participants provided written consent to the parent trial, and informed consent was waived for this retrospective study.

### CT image acquisition and coronary artery calcium assessment
All scans were performed on multi-detector (≥4) CT scanners using standard non-ECG-synchronized, non-contrast, low-dose chest CT protocols[15]. Each participant's first chest CT was considered. For 13,996/24,090 (58%) participants, CAC scores (Agatston score), calculated from a deep learning algorithm, were available as previously published[4]. The continuous CAC score was divided into four clinically relevant categories: CAC = 0; CAC = 1–100, CAC = 101–300, and CAC > 300[16].

### Epicardial adipose tissue measurement
We measured EAT volume ($cm^3$) and density (Hounsfield Units, HU) using a dedicated deep-learning algorithm in the non-contrast non-gated NLST CT data. EAT was defined as fat tissue inside the pericardial sac[14,17] (Fig. 1). Since body fat depots strongly correlated with obesity and body habitus, we indexed the absolute EAT volume by body surface area (BSA; $m^2$)[18] and used the BSA-indexed EAT volume ($cm^3/m^2$) for all analyses. Furthermore, we adjusted all multivariable models for body mass index (BMI). EAT density was defined as the average attenuation of all segmented EAT voxels within the pericardial sac.

### Deep-learning algorithm for epicardial adipose tissue segmentation.
We developed and open-sourced a deep learning system for fully automated EAT segmentation in cardiac ECG-gated and non-ECG-gated chest CT utilizing data from the large Framingham Heart Study (FHS)[19], Prospective Multicenter Imaging Study for Evaluation of Chest Pain (PROMISE)[20], and NLST[1] cohorts; this algorithm was used in a prior PROMISE analysis[12]. The deep-learning algorithm comprised three

stages: (1) heart localization, identifying and isolating/cropping the heart region from the input chest CT scans; (2) heart segmentation by identifying the pericardial sac; (3) EAT rendering within the pericardial sac (Fig. 1). EAT was defined as all voxels within the segmented pericardial sac and attenuation between −190 to −30 HU[21].

### Training and testing data sets.
A dataset of 2164 randomly selected CT scans from FHS ($n = 628$), PROMISE ($n = 1,140$), and NLST ($n = 396$) was used to develop the system. Four experienced CV radiologists provided standard manual segmentations for all 2164 cases.

We used 858 cases (FHS, $n = 628$; PROMISE, $n = 130$; NLST, $n = 100$) for training and tuning of the algorithm, while the remaining 1306 cases (PROMISE, $n = 1010$; NLST, $n = 296$) were reserved for testing.

### Deep-learning algorithm development.
The heart localization step was framed as a coarse segmentation problem on down-sampled scans and segmentation masks. We trained the heart localization model on resampled 3 mm isotropic images cropped to $112 \times 112 \times 112$ voxel cubes. The localization models predicted heart masks were used to crop the hearts by resampling them to the original voxel spacing, calculating the center of the bounding box around the mask, and finally cropping a $384 \times 384 \times 80$ voxel cube around it. We added an 11-voxel safety margin to ensure that the whole heart is captured in the cropped region.

A heart segmentation model was then trained on the cropped scans from the previous stage, with data resampled to the voxel spacing of $2.0 \times 2.0 \times 2.5$ mm and cropped to $128 \times 128 \times 80$ voxel cubes to fit the graphics processing unit (GPU) memory. Compared to the localization model, the chosen higher spatial resolution allowed for precise segmentation and improved performance.

Finally, to obtain EAT segmentations, the system automatically rendered voxels between −190 and −30 HU from the original high-resolution image in the area of the heart mask generated by the segmentation model. The same rendering step is used in manual EAT segmentation.

Both heart localization and segmentation stages used separately trained 3D U-Net architectures[22] with four down-sampling steps and a dropout rate of 0.5. To reduce the network's memory requirements, we introduced a feature reduction step in the bottleneck part of the architecture. We trained the models using Dice loss function[23] and Adam optimizer[24] with an initial learning rate of 0.0001 that we reduced by 50% (localization) or 70% (segmentation) every 100 epochs. The batch size was 3 (localization) or 4 (segmentation). Data augmentation strategies differed for the two stages. We applied a random ±10 voxels axial plane translation and ±4 degrees rotation in any axis for the localization model. We used a random ±20 voxels axial plane translation and ±35 degrees rotation in any axis for the segmentation model. Both networks were trained for 1200 epochs. While the localization stage model was trained on a single 70/30 training and tuning set split, the segmentation model used a variation of cross-validation to take advantage of all the data without overfitting. We achieved that by reselecting the training (70%) and tuning (30%), which were split randomly every 100 epochs throughout the training.

Development and evaluation were done on a Linux workstation equipped with one Nvidia RTX A6000 GPU using Tensorflow-GPU (v1.14), Keras (v2.3.1), and NVIDIA CUDA (v10.2). Using this setup, the system needed less than 2 s on average to segment EAT in a given scan. This system is the latest version of our heart segmentation system developed by our group[4,12,25,26]. Unlike the previous versions that required four GPUs, the newest algorithm can be run on single GPU or CPU-only systems.

### Deep-learning algorithm testing.
To determine the algorithm's accuracy, we compared 1306 automatically vs. manually segmented hearts in an independent data set using the original high-resolution CT images. Our system demonstrated a strong performance with a median Dice coefficient of 0.95 (Interquartile Range, IQR = 0.02; Spearman's correlation of 0.96; $p < 0.0001$), indicating a high degree of overlap between the automated and manual heart segmentations. The median surface

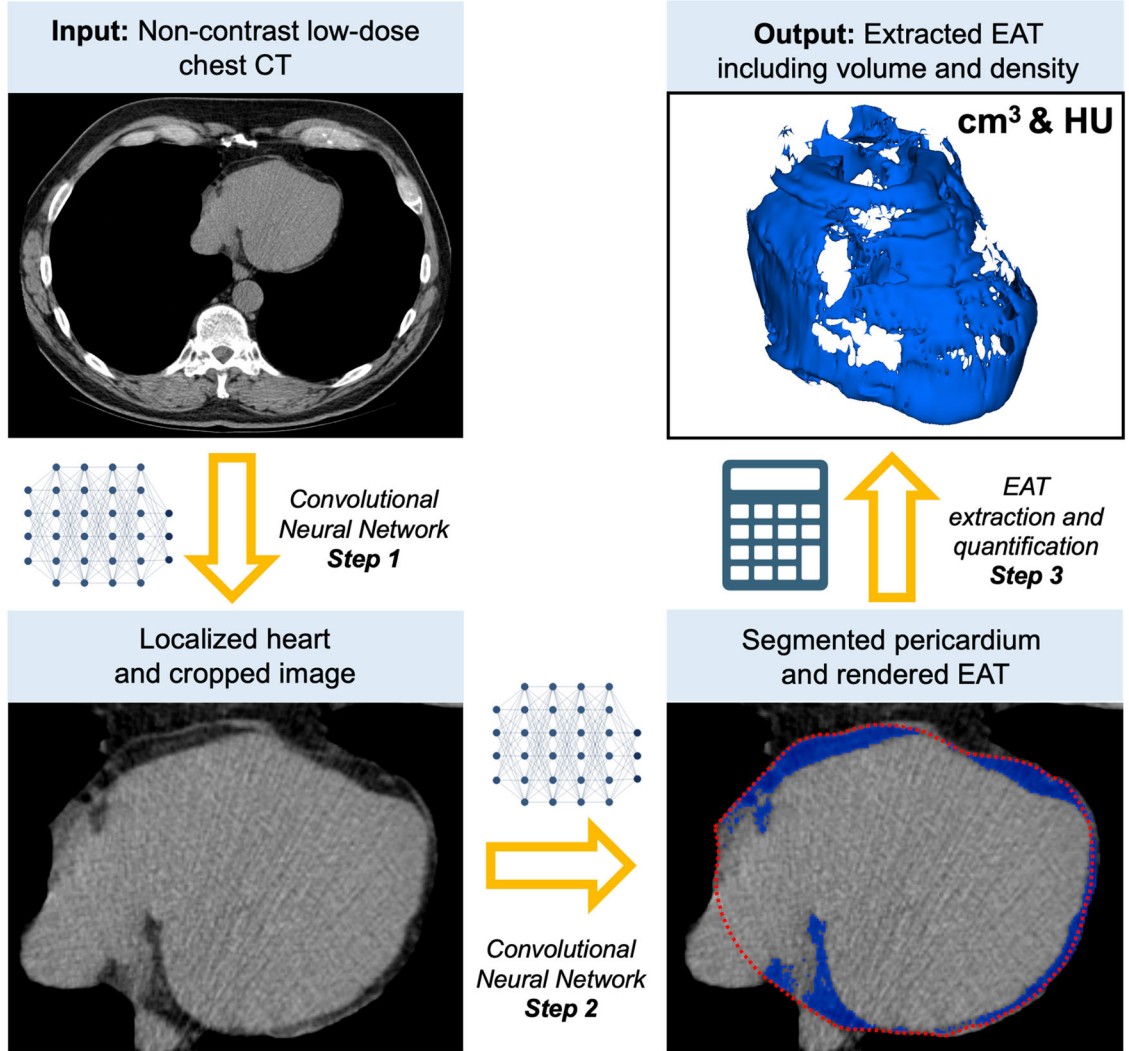

**Fig. 1 | Overview of the deep-learning framework and EAT segmentation steps implemented on lung cancer screening CT.** EAT is segmented from non-contrast low-dose chest CTs by localizing the heart, identifying the pericardium (red line), and rendering the fatty components (blue) within the pericardial sac. EAT was rendered using fixed attenuation thresholds of −190 to −30 HU. CT computed tomography, EAT epicardial adipose tissue, HU Hounsfield Units. Created with BioRender.com.

distance was 1.6 mm (IQR = 0.71), suggesting a close match between the automated and manually segmented pericardial sac. Supplementary Fig. 2 provides images of optimally and suboptimally segmented hearts.

## Statistical methods

Continuous variables were reported as mean ± SD or median (IQR), and categorical variables as absolute and relative frequencies. Clinical characteristics were stratified by survival/all-cause death, and the differences were tested with the log-rank test. We also provide EAT volume and density values across categories of clinical features, testing the differences by the Kruskal–Wallis test. Spearman's correlation was used to calculate the correlation between EAT volume and density.

Uni- and multivariable Cox proportional hazard regression models were used to calculate Hazard ratios (HR) and 95% confidence intervals (CI) to evaluate associations between the continuous EAT measures and incident all-cause and CV death. Multivariable models were adjusted for age, sex, race, ethnicity, smoking (former vs. current), number of pack years, history of heart disease, history of stroke, hypertension, diabetes, education status, BMI, and CAC score. Furthermore, we included EAT volume and density in all main multivariable regressions to test their independent prognostic value. We also performed sensitivity analyses in participants (A) without known CAD, heart failure, or stroke at baseline and (B) in participants without CAC at baseline. Cumulative event rates between EAT volume and density quartiles were compared using the Kaplan-Meier method and tested for significance using log-rank tests. We also performed C-statistic tests to assess the predictive power of the EAT measures and compared the discriminatory capacity between models using the Likelihood-ratio test for nested models. We added a cNRI analysis to assess the models' incremental value using the Stata NRI package by Lunt[27,28]. We used the continuous NRI as recommended by Pencina et al.[27,28], as continuous NRI offers the broadest and most standardized application and is not affected by different event rates and should thus be used when comparing NRIs across studies. Stata 17.1 (College Station, TX, USA) was used for all analyses, and two-sided $P$ Values of <0.05 were considered statistically significant.

## Reporting summary

Further information on research design is available in the Nature Portfolio Reporting Summary linked to this article.

## Results

### Study population

The study analyzed 24,090 NLST participants with an average age of 61 ± 5 years and 59% (14,204/24,090) of them being men. Over a median follow-up time of 12.3 (11.9–12.8) years, 19.5% (4,690/24,090) of the participants died,

with 23.2% (1,089/4,690) CV deaths. Participants who died were generally older, more likely males, African American, Non-Hispanic, active smokers with more pack years and had a history of CVD (i.e., CAD, heart failure, or stroke), more CV risk factors, higher CAC score, slightly lower BMI, and lower level of education. Table 1 details key baseline demographics and clinical characteristics.

### Association of epicardial adipose tissue with CV risk factors and CAC

The BSA-indexed mean EAT volume was $70.3 \pm 24.6$ cm³/m². It increased with age, was higher in men vs. women, and was substantially higher in Asians, followed by Caucasians and African Americans. EAT volume also

increased with rising BMI, was higher in former vs. current smokers, and increased with the number of pack years. Also, participants with prior heart disease or stroke, lower level of education, or higher CAC scores presented high EAT volumes ($P \leq 0.007$ for all; Supplementary Data 1). The association between EAT volume and CAC (*ln*-transformed CAC due to right skewness of the CAC score) remained significant in a multivariable regression analysis adjusting for age, sex, race, ethnicity, smoking status (current vs. former), number of pack years, hx of myocardial infarction, hx of stroke, diabetes mellitus, hypertension, education status, and BMI (beta per 10 cm³/m²: 0.04, 95%CI: 0.02–0.05, $P < 0.001$).

The mean EAT density was $-77.7 \pm 5.2$ HU and decreased with increasing EAT volume, reflected in a strong negative correlation (Spearman's rho: $-0.72$; $P < 0.001$; Fig. 2). Accordingly, the associations between EAT density and clinical characteristics were similar but inverse compared to EAT volume (Supplementary Data 1).

### Association of epicardial adipose tissue with mortality
**Epicardial adipose tissue volume.** EAT volume was approximately 11% higher in participants who died ($76.0 \pm 28.0$ vs. $69.0 \pm 23.5$ cm³/m²; $P < 0.001$, Table 1), and the mortality rates increased with rising EAT volume (quartiles of EAT volume—all-cause mortality: Q1 ( < 53 cm³/m²): 16.3%, Q2 (53–67 cm³/m²): 16.0%, Q3 (67–85 cm³/m²): 19.6%, Q4 (>85 cm³/m²): 25.9%; CV mortality: Q1: 3.7%, Q2: 3.3%, Q3: 4.7%, Q4: 6.4%; log-rank: $P < 0.001$ for both, Supplementary Table 1). Figure 3 provides the corresponding survival curves.

In the Cox regression analysis, an EAT volume increase by 10 cm³/m² was associated with 19% and 27% higher hazards of all-cause and CV mortality, respectively, independent of EAT density (all-cause mortality: HR = 1.19; 95%CI: 1.17–1.21; $P < 0.001$, and CV mortality: HR = 1.27; 95%CI: 1.23–1.30; $P < 0.001$). This association remained significant after further adjustment for age, sex, race, ethnicity, smoking status (current vs. former), number of pack years, history of CV disease, diabetes, hypertension, education, BMI, and categorical CAC score (HR = 1.10; 95%CI: 1.08–1.13; $P < 0.001$ and HR = 1.14; 95%CI: 1.10–1.19; $P < 0.001$, respectively) (Table 2).

**Epicardial adipose tissue volume in participants without known heart disease or CAC at baseline.** The association between EAT volume and events was significant in a subgroup of 20,454 participants without known CVD at baseline (i.e., primary prevention cohort without prior CAD, heart failure, or stroke) in the fully adjusted model (all-cause mortality: HR = 1.11; 95%CI: 1.08–1.14; $P < 0.001$, CV mortality: HR = 1.14; 95%CI: 1.08–1.20; $P < 0.001$, Supplementary Table 2).

### Table 1 | Baseline demographic and clinical characteristics stratified by all-cause death

| | All (N = 24,090) | Alive (N = 19,400) | Dead (N = 4,690) | *P* |
|---|---|---|---|---|
| **Clinical characteristics** | | | | |
| Age –years | 61 ± 5 | 61 ± 5 | 64 ± 6 | <0.001 |
| Sex –male | 14,204 (59.0) | 11,056 (57.0) | 3,148 (67.1) | <0.001 |
| Race | | | | <0.001 |
| Caucasian | 21,950 (91.1) | 17.718 (91.3) | 4,232 (90.2) | |
| African American | 1,050 (4.4) | 795 (4.1) | 255 (5.4) | |
| Asian | 524 (2.2) | 447 (2.3) | 77 (1.6) | |
| Other/Unknown | 566 (2.4) | 440 (2.3) | 126 (2.7) | |
| Ethnicity | | | | 0.001 |
| Hispanic/Latinx | 429 (1.8) | 367 (1.9) | 62 (1.3) | |
| Non-Hispanic/Latinx | 23,559 (97.8) | 18,963 (97.7) | 4,597 (98.0) | |
| Other/Unknown | 102 (0.4) | 71 (0.4) | 31 (0.7) | |
| Smoking | | | | <0.001 |
| Former | 12,518 (52.0) | 10,558 (54.4) | 1,960 (41.8) | |
| Current | 11,572 (48.0) | 8,842 (45.6) | 2,730 (58.2) | |
| Pack years | 48.0 (39.0–66.0) | 46.5 (39.0–64.5) | 55.0 (43.0–78.0) | <0.001 |
| Hx of heart disease of MI | 3.095 (12.88) | 2,110 (10.9) | 985 (21.1) | <0.001 |
| Hx of stroke | 660 (2.7) | 419 (2.2) | 241 (5.2) | <0.001 |
| Diabetes mellitus | 2,326 (9.7) | 1,576 (8.1) | 750 (16.0) | <0.001 |
| Hypertension | 8,462 (35.2) | 6,444 (33.3) | 2,018 (43.1) | <0.001 |
| Level of education | | | | <0.001 |
| High school graduate or below | 7,025 (29.2) | 5,343 (27.5) | 1,682 (35.9) | |
| Post high school (excluding college) | 3,389 (14.1) | 2,703 (13.9) | 686 (14.6) | |
| Some college or bachelor's degree | 9,715 (40.3) | 8,027 (41.4) | 1,688 (36.0) | |
| Graduate school | 3,479 (14.4) | 2,942 (15.2) | 537 (11.5) | |
| Other/unknown | 482 (2.0) | 385 (2.0) | 97 (2.1) | |
| BMI –kg/m² | 27.9 ± 5.0 | 27.9 ± 4.9 | 27.7 ± 5.5 | <0.001 |
| BSA –m² | 1.99 ± 0.3 | 1.99 ± 0.25 | 2.00 ± 0.26 | 0.003 |
| **CT measures** | | | | |
| EAT volume (BSA-indexed) –cm³/m² | 70.3 ± 24.6 | 69.0 ± 23.5 | 76.0 ± 28.0 | <0.001 |
| EAT density –HU | −77.7 ± 5.2 | −77.6 ± 5.1 | −78.0 ± 5.5 | <0.001 |
| CAC score* | 61.8 (1.3–375.5) | 41.6 (0–294.2) | 202.9 (20.9–786.6) | <0.001 |

Unless otherwise specified, data are numbers of participants, with percentages in parentheses, means ± standard deviation, or median (Q1–Q3). *CAC score was available in a subgroup of 13,966 participants. BMI = body mass index, BSA = body surface area, CAC = coronary artery calcium, EAT = epicardial adipose tissue, HU = Hounsfield Units, MI = myocardial infarction.

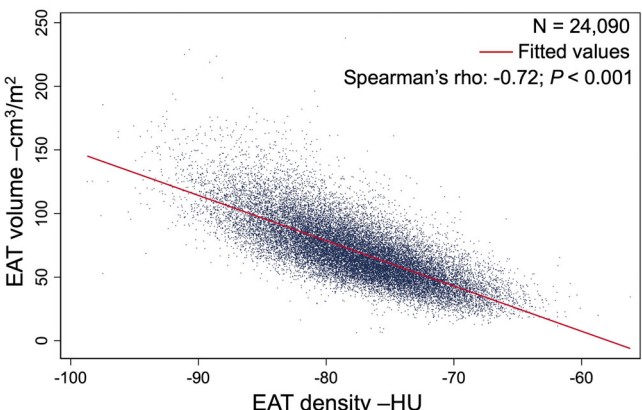

**Fig. 2 | Association between EAT volume and density.** The scatterplot shows a strong negative correlation between EAT volume and density, as shown by the fitted red regression line. Blue dots represent individual observations. EAT = epicardial adipose tissue, HU = Hounsfield Units.

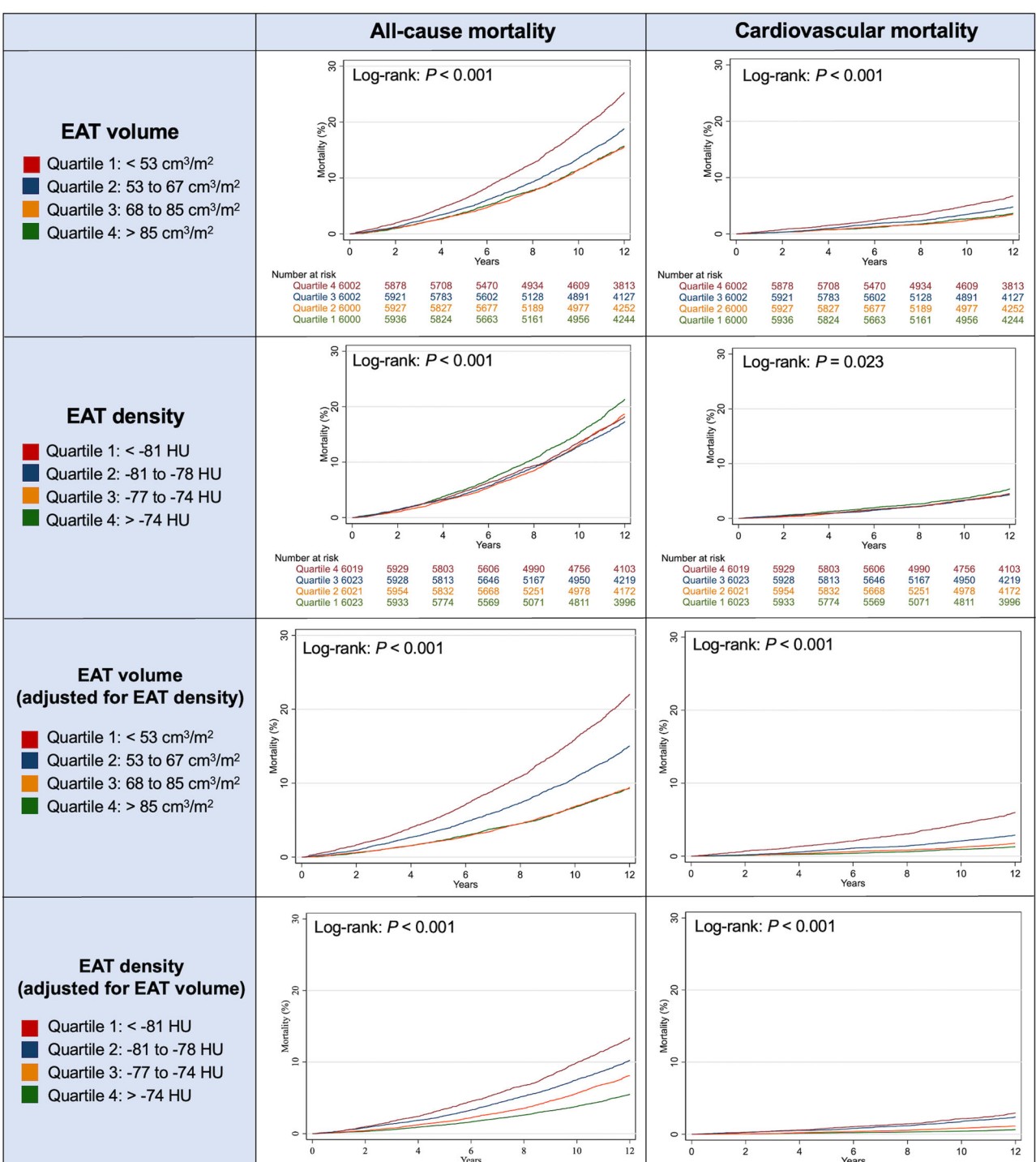

**Fig. 3 | Kaplan-Meier survival analysis of EAT volume and density for all-cause and cardiovascular mortality.** A two-sided log-rank test was used to calculate the P values. EAT = epicardial adipose tissue, HU = Hounsfield Units.

In a subgroup of participants without CAC (3,355/13,966, 24.0%), EAT volume was also related to all-cause and CV mortality in the fully adjusted analyses (all-cause mortality: HR = 1.10; 95%CI: 1.03–1.17; P = 0.003, CV mortality: HR = 1.21; 95%CI: 1.05–1.40; P = 0.01; Supplementary Table 3).

**Epicardial adipose tissue density.** Overall, EAT density was −77.7 ± 5.2 HU and slightly lower in those who died vs. survivors (−78.0 ± 5.5 vs. −77.6 ± 5.1 HU; P < 0.001), reflected in a negative univariable association between EAT density and adverse events (all-cause mortality (per 10 HU) HR: 0.87, 95%CI: 0.82–0.92, P < 0.001;

CV mortality (per 10 HU) HR: 0.88, 95%CI: 0.79–0.99, p = 0.032). Nevertheless, in a multivariable regression analysis accounting for EAT volume, an increase of EAT density by 10 HU was associated with a 66% higher hazard of all-cause mortality and more than a twofold risk in CV mortality (all-cause mortality: HR = 1.66; 95%CI: 1.53–1.80; P < 0.001, CV mortality: HR = 2.14; 95%CI: 1.81–2.52; P < 0.001). This association remained significant after additional adjustment for age, sex, race, ethnicity, smoking status (current vs. former), number of pack years, history of CV disease, diabetes, hypertension, education status, BMI, and categorical CAC score (all-

**Table 2 | Association of EAT volume and density with all-cause and cardiovascular mortality**

| | Univariable | | | Model 1 | | | Model 2 | | | Model 3 | | |
|---|---|---|---|---|---|---|---|---|---|---|---|---|
| All-cause mortality | HR | 95%CI | P | HR | 95%CI | P | HR | 95%CI | P | HR | 95%CI | P |
| EAT volume –cm³/m² | 1.10 | 1.09–1.12 | <0.001 | 1.19 | 1.17–1.21 | <0.001 | 1.10 | 1.08–1.12 | <0.001 | 1.10 | 1.08–1.13 | <0.001 |
| EAT density –HU | 0.87 | 0.83–0.92 | <0.001 | 1.66 | 1.53–1.80 | <0.001 | 1.36 | 1.26–1.48 | <0.001 | 1.38 | 1.24–1.54 | <0.001 |
| Cardiovascular mortality | | | | | | | | | | | | |
| EAT volume –cm³/m² | 1.14 | 1.11–1.16 | <0.001 | 1.27 | 1.23–1.30 | <0.001 | 1.15 | 1.11–1.19 | <0.001 | 1.14 | 1.10–1.19 | <0.001 |
| EAT density –HU | 0.88 | 0.79–0.99 | 0.037 | 2.14 | 1.81–2.52 | <0.001 | 1.76 | 1.49–2.09 | <0.001 | 1.78 | 1.42–2.22 | <0.001 |

All hazard ratios (HR) are per 10 cm³/m² or 10 HU change.
Model 1: EAT volume and density.
Model 2: Model 1 + age, sex, Race, Ethnicity, smoking (current vs. former), pack-years, history of heart disease, history of stroke, diabetes mellitus, hypertension, education status, and BMI.
Model 3: Model 2 + CAC score. BMI = body mass index, CAC = coronary artery calcium, EAT = epicardial adipose tissue, HU = Hounsfield Units. Full models are provided in Supplementary Table 4.

**Table 3 | Incremental prognostic value of EAT volume and density**

| | Harrell's C concordance statistics | Likelihood-ratio test (Chi²) | P | cNRI Improvement | P |
|---|---|---|---|---|---|
| All-cause mortality | | | | | |
| Clinical parameters* | 0.700 | Ref. | - | Ref. | - |
| CAC score | 0.611 | - | - | - | - |
| EAT† | 0.579 | - | - | - | - |
| Clinical parameters + CAC score | 0.706 | 86.8 | <0.001 | 0.27 | <0.001 |
| Clinical parameters + EAT | 0.705 | 72.4 | <0.001 | 0.12 | <0.001 |
| Clinical parameters + CAC score + EAT | 0.710 | 159.9 | <0.001 | 0.27 | <0.001 |
| Cardiovascular mortality | | | | | |
| Clinical parameters* | 0.751 | Ref. | - | Ref. | - |
| CAC score | 0.672 | - | - | - | - |
| EAT† | 0.625 | - | - | - | - |
| Clinical parameters + CAC score | 0.769 | 72.7 | <0.001 | 0.49 | <0.001 |
| Clinical parameters + EAT | 0.759 | 31.0 | <0.001 | 0.16 | <0.001 |
| Clinical parameters + CAC score + EAT | 0.776 | 104.2 | <0.001 | 0.49 | <0.001 |

*Clinical parameters function as a reference group for the improvement calculation, including age, sex, Race, Ethnicity, smoking (former vs. current), number of pack years, history of heart disease, history of stroke, hypertension, diabetes, education status, and BMI.
†EAT volume and density. All analyses were based on a random subset of 13,860 individuals with complete CAC data. CAC = coronary artery calcium, cNRI = continuous net reclassification index, CV = cardiovascular, EAT = epicardial adipose tissue.

cause death: HR = 1.38; 95%CI: 1.24–1.54; P < 0.001, CV death: HR = 1.78; 95%CI: 1.42–2.22; P < 0.001; Table 2).

**Epicardial adipose tissue density in those without prior heart disease or CAC at baseline.** Supplemental subgroup analyses in participants without known CV disease revealed a strong and independent association between EAT density and events (Supplementary Table 2). In participants without CAC (n = 3,355), EAT density was associated with CV mortality in a model adjusted for EAT volume (HR = 2.05; 95%CI: 1.03–4.09; P = 0.042). In addition, there was a trend towards an association between EAT density and CV death in the fully adjusted model correcting for CV risk factors, education, EAT volume, and BMI (HR = 1.83; 95%CI: 0.90–3.71; P = 0.096; Supplementary Table 3). Of note, there were only 63 CV deaths in the relatively low-risk subgroup with CAC = 0, and the analysis may have been underpowered.

**Incremental value of epicardial adipose tissue volume and density**
The combination of clinical parameters with CAC and EAT was most predictive of CV mortality (Harrel's C concordance: 0.78), while being less predictive of all-cause mortality (Harrel's C concordance: 0.71). Adding EAT to clinical risk factors and clinical risk factors + CAC score increased the discriminatory capacity significantly from 0.70 to 0.71 (all-cause mortality) and 0.75 to 0.78 (CV-mortality) (P < 0.001 for all). The continuous net reclassification improvement (cNRI) test revealed that adding EAT measures to CV risk factors improved the prediction of all-cause (cNRI: 0.12; P < 0.001) and CV death (cNRI: 0.16; P < 0.001). We found the greatest improvement when adding both EAT and CAC to clinical CV risk factors with cNRI of 0.27 and 0.49 for all-cause and CV mortality, respectively (both P < 0.001) (Table 3).

## Discussion
People undergoing lung cancer screening are at high CV risk. However, up until now, the full potential of routine low-dose chest CTs has not been fully utilized to determine a person's mortality risk. Our study demonstrates that an automated deep learning-based epicardial fat quantification accurately stratifies the risk for all-cause and CV mortality across 24,090 heavy smokers enrolled in the NLST. In addition, our results demonstrate that automated EAT assessment is a feasible and reliable way of predicting mortality independent of

and incremental to persons' demographics, clinical characteristics, CV risk factors, body habitus, and CAC score.

Heavy smokers are at increased CVD risk and need better risk stratification. Multiple large clinical trials have shown that low-dose chest CT is an effective screening method for lung cancer, leading to reduced lung cancer and all-cause mortality[1–3,29,30]. However, these trials also revealed that more screened individuals died of CV causes than cancer. For example, in NLST, 356 participants died of lung cancer over six years of follow-up, while 486 died of CV disease[1]. The recent Nederlands–Leuvens Longkanker Screenings Onderzoek (NELSON) study indicated the effectiveness of low-dose chest CT for lung cancer screening with a cancer-related mortality rate decrease but similarly high CV mortality compared to NLST, emphasizing the importance of CV risk estimation to reduce overall mortality[2]. CAC scoring can be measured on lung cancer screening CTs and reclassifies CV risk in heavy smokers[4]. However, a study of 3,110 individuals showed that despite significant CAC being a frequent finding (26%) on lung-screening CT, only a small portion (31%) of those with CAC had been diagnosed with CAD before the CT[31]. These results emphasize the unmet need for improved CV risk stratification and the opportunity to leverage CT images in this high-risk population better. Our results indicate that EAT provides prognostic value beyond CAC scoring and is associated with mortality even in people without CAC.

Lung cancer and CVD have overlapping clinical risk factors such as obesity, diabetes mellitus, hypertension, tobacco smoking, poor diet, and sedentary lifestyle[32]. Several of these risk factors, including smoking status, diabetes, hypertension, and obesity, were associated with higher EAT volume and density in our study. These findings suggest that changes in EAT might be part of a shared mechanistic pathway for both cancer and CVD.

While inflammation is a well-established pathway that drives cancer and atherogenesis[32,33], EAT has been tied to local perivascular inflammation. EAT, a metabolically active visceral fat depot, encases the coronary arteries directly without a basal membrane functioning as a barrier. This arrangement facilitates a direct exchange of inflammatory mediators and promotes atherogenesis[6,7]. For instance, in a study of patients with critical CAD who received coronary artery bypass grafts, harvested EAT samples showed higher levels of proinflammatory IL-1β, IL-6, TNFα, and MCP-1 than paired subcutaneous fat controls[7]. EAT inflammation can be measured on CT, because inflamed EAT changes its morphology through increased lipolysis, inhibited lipogenesis, and increased perivascular edema, resulting in a lower lipid-to-connective tissue and water ratio and a higher EAT density on CT[6,34]. A recent substudy from the Early Identification of Subclinical Atherosclerosis by Non-invasive Imaging Research (EISNER) study reported correlations of EAT volume and density with systemic inflammatory biomarkers (e.g., CRP, IL-6, PAI-1, and MMP-9)[10]. Our findings support these mechanistic insights, indicating that inflammation, represented by EAT volume and density alterations, may hold prognostic significance in heavy smokers.

Our study corroborates growing evidence linking increased EAT volume and density with adverse events. For instance, a large meta-analysis including over 20,000 individuals (mainly from the FHS, Multi-Ethnic Study of Atherosclerosis, Heinz Nixdorf Recall Study, EISNER study, and the Rotterdam study) reported a strong relationship between EAT volume, CV risk factors, and CAD severity[35]. Furthermore, Rajani et al. reported an increased EAT volume in patients with high-risk plaque phenotype beyond traditional CV risk (Odds ratio 1.7, $P = 0.04$)[36], results also seen in a large cohort of stable chest pain patients in the PROMISE study[12]. In our study, EAT was associated with adverse events incremental to CAC score and in a subgroup of participants without CAC or known CVD at baseline (i.e., primary prevention cohort). These results show that EAT, as an index of inflammation, may be a precursor of CAD rather than just an imaging marker of prevalent disease. To the best of our knowledge, our study is the first to show prognostic value of EAT volume and density in a large prospectively collected sample of heavy smokers, a group at markedly increased CV risk. Therefore, increased EAT might be a valuable new parameter for identifying high-risk patients in this vulnerable cohort requiring preventive care.

There is an unmet need for automated EAT measurement tools. EAT assessment is relatively new and usually requires ECG-synchronized cardiac CT images, specialized software, and manual measurement by expert readers. Consequently, EAT volume and density are not routinely reported on non-cardiac chest CTs. We address this issue and provide a robust and accurate automated algorithm to assess EAT volume and density. In addition, the algorithm offers EAT measurements in under 2 s without human input, making it an "end-to-end" solution for CV risk assessment in clinical settings. Although other research groups have developed deep learning algorithms for automated EAT quantification[10,37–39], these groups used only dedicated ECG-gated cardiac CT scans, smaller cohorts, or proprietary technologies.

Our deep learning algorithm has several strengths. First, it has been developed in collaboration with a core laboratory that assessed EAT in a standardized fashion using data from multiple large, well-phenotyped cohorts and randomized controlled trials (e.g., FHS, PROMISE). Finally, we share the deep-learning system with the community and believe that sharing the algorithm will contribute to scientific advancement, foster collaboration, and accelerate the adoption of this new technology by academic and commercial entities.

We acknowledge several limitations of the current study. First, some standard CV risk factors (e.g., lipids) were not recorded in the NLST trial, and adjustment for the standard 10-year atherosclerotic CVD risk was not possible. However, most of the strongest predictors (e.g., age and sex) were available and included in the study among other variables influencing outcomes and EAT values (e.g., BMI). Second, nonfatal CV events (e.g., nonfatal myocardial infarction or stroke) were not recorded; hence, our study provides data only on all-cause and CV mortality. Third, the NLST cohort included only heavy smokers (≥30 pack years), and differences in EAT's prognostic value could not be calculated between heavy smokers, non-smokers, and lower-risk smokers with ≥20–30 pack years, a group currently also recommended for lung cancer screening. Future work is necessary to understand the implications for non-smokers and lesser-smoking populations. Furthermore, CAC scores were available in a representative random sample (N = 13,966) of the NLST data set. Finally, most NLST participants were white non-Hispanics; thus, future studies in more diverse populations are needed.

## Conclusions

EAT volume and density are independently associated with all-cause and CV mortality in the high-risk group of lung cancer screening participants. Deep-learning-based EAT assessment stratifies mortality risk beyond traditional CV risk factors and CAC score and, therefore, may guide primary CVD prevention in heavy smokers and improve their outcomes. EAT assessment is a unique method that serves a large group of high-risk individuals without a need for additional imaging or manual measurements, avoiding radiation while saving time and costs. Future prospective studies are warranted to investigate the impact of adding EAT measurements to routine lung cancer screening reports.

## Data availability

The National Cancer Institute (NCI) forbids sharing the NLST data with third parties; however, the main NLST data set, including raw CT images, is available by application directly through the NCI and can be requested at https://biometry.nci.nih.gov/cdas/nlst. Source data which refers to the numerical values underlying Figs. 2, 3 in this article are available via Supplementary Data 2.

## Code availability

The underlying heart segmentation system we utilized to quantify epicardial adipose tissue can be assessed via GitHub at https://github.com/AIM-Harvard/DeepHeart or under https://zenodo.org/records/10724420[40].

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

## Acknowledgements
We thank the NLST participants and investigators for the data collection. Original data collection for the ACRIN 6654 trial (NLST) was supported by National Cancer Institute (NCI) Cancer Imaging Program grants. The statements contained herein are solely those of the authors and do not represent or imply concurrence or endorsements by the above organizations.

## Author contributions
B.F. conceptualized the study, analyzed and interpreted the data, and wrote the manuscript. I.H. and R.Z. performed the deep-learning segmentations. M.C.L. supported manuscript writing and data analysis. V.K.R. supported the deep-learning analysis and provided quality control. T.M. supervised the statistical analysis. M.T.L. supervised the analysis and was a contributor to writing the manuscript. J.H.W.L.A. oversaw deep-learning system development and implementation, data curation, and analysis. All authors read and approved the final manuscript.

## Competing interests
The authors declare competing interests.

## Additional information
**Supplementary information** The online version contains Supplementary Material available at https://doi.org/10.1038/s43856-024-00475-1.

