## [Peer Review File · Communications Medicine]

Reviewers' comments:

Reviewer #1 (Remarks to the Author):

The authors investigated the prognostic value of deep learning-based automated epicardial adipose tissue (EAT) quantification and compared it to traditional clinical risk factors and coronary artery calcium (CAC) scoring. The study analyzed 24,090 National Lung Screening Trial (NLST) participants who underwent non-ECG-synchronized, non-contrast low-dose chest CT. During the follow-up period, EAT volume and density were independently associated with all-cause and cardiovascular mortality. EAT quantification provided incremental prognostic value beyond clinical risk factors, body mass index (BMI), education, and CAC score. These findings suggest that EAT volume and density measurements from low-dose chest CT are independent predictors of all-cause and cardiovascular mortality in heavy smokers, with potential implications for CVD risk stratification in this high-risk population.

While the clinical utility would be suggested and I think this research is suitable for Communication Medicine, I have several concerns.

1. The authors improved their automated EAT measurement tool according to the supplementary method. (The current version is three)

On the other hand, similar algorithms were found in previous reports (doi: 10.1109/TMI.2018.2804799, <https://doi.org/10.1038/s41598-022-06351-z>).

How was the performance improvement of the authors' algorithm compared to previous ones?

2. A article with a similar concept was published regarding low-dose chest CT and cardiovascular disease, in which Not measuring EAT but assessing chest CT predicted cardiovascular disease risk (<https://doi.org/10.1038/s41467-021-23235-4>).

What are the superior points to focus on EAT measurements compared to the above?

3. EAT volume and EAT density were closely correlated according to Fig.2.

So, I suppose one of them could be enough to predict prognosis. Please discuss and explain the rationale to include both EAT volume and EAT density.

4. The result was concluded just from one dataset. Could the authors validate and replicate it using other cohorts like LIDC-IDRI and NSCLC Radiogenomics?

5. Is the conclusions just applied to groups of heavy smokers? Please discuss the situations in non-smoking and mild-smoking groups.

Reviewer #2 (Remarks to the Author):

Please see attachment for a formatted reievew.

This manuscript shows that EAT volume and density from lung cancer screening images are associated with cardiovascular and all cause mortality. Effects are beyond clinical factors, BMI, and Agatston score. This is a significant contribution towards the use of chest CT images to obtain improved cardiovascular prediction. The manuscript needs considerable editing and additional

analyses as it was difficult to follow.

Some specific concerns are:

- There must be a better description of EAT segmentation. This is my biggest concern as all else flows from this. In the main document, it is not clear that this is the paper with “validation” of the method. An assertion was made with references 14 and 34. Neither one supports this claim. This led me on a wild goose chase trying to track down the validation. Then I found some validation in the Supplement. I had issues with the meager text.

- o I could not understand the text below.

This system, used in the present study and in (4), is the third version of our heart segmentation system. The first (5) and second (6) versions were trained on 129 and 858 CT scans, respectively, and achieved median Dice coefficients of 0.90 and 0.95, respectively

- o What is the loss function?

- o As this is the validation for the entire result, I would have expected some figures showing good segmentations and bad segmentations.

- o With the extensive down sampling, I am surprised that one can even see the pericardial sac. Is this a magic down-sampling technique?

- o There is a lot of talk about versions and data. It is impenetrable. It probably makes perfect sense, but I do not understand it.

- o Was testing indeed done on a held-out test set of 1306 scans, all of which were volumetrically manually labeled?

- o Was testing done at the original resolution or down-sampled resolution? If the original, a DICE of 0.95 hardly makes sense as the associated partial volume errors would probably give a worse DICE than this.

- o What is the loss function?

- o It would be nice to compare EAT from lung cancer screening to a gated exam.

- o Probably more is missing.

- As some scans will suffer from motion making it very difficult to identify the pericardial sac, I am surprised that authors did not simply lump EAT and paricardial adipose.

- For that matter, is there any association of risk with paricardial adipose?

- Table 3. What is Model 3?

- Table 4. How much of the incremental CAC value is due to the power of zero?

- Results. What is this? 10 cm³/m². At first, I thought it was normalized by height. But that does not make sense. Please explicitly describe.

- In some analyses, the EAT and HU HRs for cardiovascular death was higher than for all cause. Yet, when we look at Fig 3, there is a remarkable effect of EAT and HU on all cause! Are there any inconsistencies here? I would like to see some discussion on Fig 3 and all-cause death.

Reviewer #3 (Remarks to the Author):

This research study investigated the prognostic value of deep learning-based automated epicardial adipose tissue (EAT) quantification in heavy smokers at risk for cardiovascular disease (CVD). The study compared the predictive capabilities of EAT quantification with traditional clinical risk factors and coronary artery calcium (CAC) scoring. The researchers analyzed data from 24,090 participants in the National Lung Screening Trial (NLST). These participants underwent non-ECG-synchronized,

non-contrast low-dose chest computed tomography (CT) scans. A previously developed deep-learning algorithm (for ECG-gated CT) was employed to quantify the volume and density of epicardial adipose tissue (EAT).

Over a median follow-up period of 12.3 years, EAT volume and density were independently associated with all-cause and cardiovascular mortality. This suggests that assessing EAT volume and density through low-dose chest CT scans could be a useful tool for CVD risk stratification in heavy smokers, who are already at an increased risk for cardiovascular problems due to their smoking habits. The findings of this research study suggest that incorporating EAT quantification into routine lung cancer screening CT scans can provide valuable information for assessing the risk of cardiovascular disease and mortality in heavy smokers.

While the overall concept is worthwhile and the NLST cohort is appropriate for this study, there are several significant limitations, which need to be addressed, mostly related to the methods used and to the patient selection within the NLST cohort.

A major and critical fault is that the EAT probability score is based on machine learning (even if it is a simple logistic regression) and is designed to maximize the mortality prediction, but it was developed in the same population it is subsequently tested in. This is likely causing a significant overfit. Importantly, this probability score is shown to be the key variable, but development and testing in the same cohort undermine its significance.

Moreover, the marginal incremental AUC gain of 1% by adding EAT score for mortality prediction (is it significant by AUC ?) may be attributed from the leakage of the outcome data when fitting the EAT probability score.

The overall performance of clinical parameters alone is better than any of the imaging parameters.

It is not clear that in Table 4 the clinical+CAC+EAT is better than clinical+CAC. Importantly EAT already includes information about mortality from the fitting as described in the appendix.

Why is the CAC score available on the subset only? Was the additional value of EAT compared on 13996/24090 images only? How was this subset selected? Previously the same group published a study in nature communications where 14,959 cases were processed for CAC scoring. How were the CAC scores obtained for this analysis? Seems many different sub-cohorts are utilized.

Were stents and pacemakers excluded from the CAC analysis?

Since EAT volume and attenuation are negatively (and significantly) correlated, they shouldn't be in the same model. The authors do not present EAT attenuation in a model without EAT volume. EAT attenuation and volume should be presented separately.

There is no validation of quantification of EAT for nongated scans provided.

The presented NRI is continuous and does not have a straightforward clinical interpretation. Can categorical positive/negative NRIs and 95% CI be provided?

Why is CAC modeled as a categorical variable, but EAT is modeled as continuous? Why?

Author affiliations missing for MCL, VKR, TM

Table 3 - model 3 definition is missing (which covariates were adjusted?)

Why is the distinction of CAC categories <300 HU (standard to have 400 HU as threshold)- misleading?

For Table 4, what is the N value?

In addition, the algorithm offers EAT measurements in under 2 seconds without human input, making it an “end-to-end” solution for CV risk assessment in clinical settings” -On what kind of a computer? Is this GPU enabled, and what kind of GPU? Does it include CAC scoring?

Discussion - Inflammation is a common pathophysiology pathway for CVD and cancer -this paragraph is speculative; no data is presented in the manuscript relating to this

Same with the Epicardial adipose tissue section may be an index and promotor of local coronary artery inflammation. This paragraph is unrelated to the paper.

Deep Learning Analysis of Epicardial Adipose Tissue on Non-ECG-gated Low-Dose Chest CT: Uncovering Hidden Cardiovascular Risk in Heavy Smokers

Foldyna

This manuscript shows that EAT volume and density from lung cancer screening images are associated with cardiovascular and all cause mortality. Effects are beyond clinical factors, BMI, and Agatston score. This is a significant contribution towards the use of chest CT images to obtain improved cardiovascular prediction. The manuscript needs considerable editing and additional analyses as it was difficult to follow.

Some specific concerns are:

- There must be a better description of EAT segmentation. This is my biggest concern as all else flows from this. In the main document, it is not clear that this is the paper with “validation” of the method. An assertion was made with references 14 and 34. Neither one supports this claim. This led me on a wild goose chase trying to track down the validation. Then I found some validation in the Supplement. I had issues with the meager text.
 - I could not understand the text below.

This system, used in the present study and in (4), is the third version of our heart segmentation system. The first (5) and second (6) versions were trained on 129 and 858 CT scans, respectively, and achieved median Dice coefficients of 0.90 and 0.95, respectively
 - What is the loss function?
 - As this is the validation for the entire result, I would have expected some figures showing good segmentations and bad segmentations.
 - With the extensive down sampling, I am surprised that one can even see the pericardial sac. Is this a magic down-sampling technique?
 - There is a lot of talk about versions and data. It is impenetrable. It probably makes perfect sense, but I do not understand it.
 - Was testing indeed done on a held-out test set of 1306 scans, all of which were volumetrically manually labeled?
 - Was testing done at the original resolution or down-sampled resolution? If the original, a DICE of 0.95 hardly makes sense as the associated partial volume errors would probably give a worse DICE than this.
 - What is the loss function?
 - It would be nice to compare EAT from lung cancer screening to a gated exam.
 - Probably more is missing.
- As some scans will suffer from motion making it very difficult to identify the pericardial sac, I am surprised that authors did not simply lump EAT and paricardial adipose.
- For that matter, is there any association of risk with paricardial adipose?
- Table 3. What is Model 3?
- Table 4. How much of the incremental CAC value is due to the power of zero?
- Results. What is this? $10 \text{ cm}^3/\text{m}^2$. At first, I thought it was normalized by height. But that does not make sense. Please explicitly describe.
- In some analyses, the EAT and HU HRs for cardiovascular death was higher than for all cause. Yet, when we look at Fig 3, there is a remarkable effect of EAT and HU on all cause!

Are there any inconsistencies here? I would like to see some discussion on Fig 3 and all-cause death.

Point-by-point responses to the editor and reviewers

We thank the editor and the reviewers for their valuable feedback and insightful comments and appreciate the time and effort they have put into reviewing our manuscript. We agree with the suggestions and made the necessary modifications for the manuscript's improvement. Please find detailed responses to individual comments below:

Referee #1:

The authors investigated the prognostic value of deep learning-based automated epicardial adipose tissue (EAT) quantification and compared it to traditional clinical risk factors and coronary artery calcium (CAC) scoring. The study analyzed 24,090 National Lung Screening Trial (NLST) participants who underwent non-ECG-synchronized, non-contrast low-dose chest CT. During the follow-up period, EAT volume and density were independently associated with all-cause and cardiovascular mortality. EAT quantification provided incremental prognostic value beyond clinical risk factors, body mass index (BMI), education, and CAC score. These findings suggest that EAT volume and density measurements from low-dose chest CT are independent predictors of all-cause and cardiovascular mortality in heavy smokers, with potential implications for CVD risk stratification in this high-risk population.

While the clinical utility would be suggested and I think this research is suitable for Communication Medicine, I have several concerns.

- 1. The authors improved their automated EAT measurement tool according to the supplementary method. (The current version is three). On the other hand, similar algorithms were found in previous reports (doi: 10.1109/TMI.2018.2804799, <https://doi.org/10.1038/s41598-022-06351-z>). How was the performance improvement of the authors' algorithm compared to previous ones?*

We thank the reviewer for their insightful comments and the opportunity to clarify our work in relation to the referenced studies (1) and (2). While these studies highlight the efficacy of their respective automatic EAT segmentation tools through comparison with manual measurements, our research primarily centers on the prognostic implications of such an automated approach, contrasting it with conventional clinical risk factors. Consequently, the EAT segmentation tool we developed serves a subsidiary role in our study.

Nonetheless, it is worth noting that our segmentation tool was conceived and validated using a significantly larger and more diverse dataset compared to studies (1) and (2). Specifically, study (1) utilized a total of 89 scans, study (2) employed 250 scans, whereas our research incorporated a total of 2,164 manually labeled cardiac and non-cardiac CT scans across three distinct cohorts of asymptomatic and stable chest pain patients as well as people undergoing lung cancer screening. Moreover, our segmentation tool is openly accessible to the public.

It is worth mentioning that our study focuses on metrics derived from heart segmentations, while studies (1) and (2) concentrate on EAT segmentations. Both methods are equally viable – the masks will only differ in the pericardium area. At the same time, the remaining voxels will be equal between manual and automatic EAT segmentation due to HU thresholding. This comparative information has been included in our paper, along with appropriate citations to these studies.

- <https://doi.org/10.1038/s41598-022-06351-z>
- <https://www.ncbi.nlm.nih.gov/pmc/articles/PMC6076348/pdf/nihms943682.pdf>

2. *A article with a similar concept was published regarding low-dose chest CT and cardiovascular disease, in which Not measuring EAT but assessing chest CT predicted cardiovascular disease risk (<https://doi.org/10.1038/s41467-021-23235-4>). What are the superior points to focus on EAT measurements compared to the above?*

Thank you for bringing the Chao et al. study to our attention. The referenced paper (1) presents a deep learning (DL) model that extracts features from the heart in three views, combines them, and uses them to classify cardiovascular disease (CVD) risk. The study team validated the DL-based CVD mortality prediction model against standard coronary CTA-derived imaging markers (CAC score and CAD-RADS class) and the clinical 10-year MESA risk score. This study chose a less supervised approach and developed an imaged-based novel risk stratification system. This system performs well in the derivation and validation cohorts, but due to its design, it delivers limited insights into coronary artery disease pathophysiology. Furthermore, the study does not provide nested models that show the incremental value of the prediction model to established markers and traditional cardiovascular (CV) risk.

We were more interested in applying our DL algorithm as a tool to assess EAT, a known imaging marker of CV risk, in which biology is well understood. Still, its prognostic value has not been investigated in a primary prevention cohort of people who undergo lung cancer screening. EAT is a metabolically active tissue, and its volume and density are closely related to vascular inflammation and atherogenesis. Therefore, changes in EAT may be precursors of coronary artery disease rather than surrogate markers of existing disease (e.g., CAC score) and, therefore, may have incremental prognostic value, specifically in primary prevention cohorts. The EAT segmentations produced are tangible and can be directly compared, offering a clear understanding of the process and results.

In contrast, a deep learning CVD risk estimation can be considered a "black box" approach, where the internal workings are not directly interpretable. While both methods are valid, they are distinctly different. The simplicity and transparency of EAT segmentation make it easier to adopt and understand. The essential advantage of our approach lies in its transparency.

1. <https://doi.org/10.1038/s41467-021-23235-4>

3. *EAT volume and EAT density were closely correlated according to Fig.2. So, I suppose one of them could be enough to predict prognosis. Please discuss and explain the rationale to include both EAT volume and EAT density.*

Thank you for pointing out the individual EAT measures. Higher EAT volume was indeed associated with events (raw EAT volume (per 10 cm³) HR: 1.04, 95%CI: 1.04–1.05, p<0.001, BSA-indexed EAT volume (per 10 cm³/m²) HR: 1.10, 95%CI: 1.09–1.12, p<0.001).

EAT density was also associated with events but with an HR <1 (EAT density (per 10 HU) HR: 0.87, 95%CI: 0.82–0.92, p<0.001). The EAT volume confounds the raw EAT density value, as shown by the strong negative correlation between EAT volume and density as shown in **Figure 2**. (rho: -0.72, p<0.001). In other words, higher EAT volume automatically leads to lower (more negative) EAT density. From the biological perspective, inflammation of EAT is known to increase EAT density by increasing water content (i.e., edema) and connective tissue to lipid ratio. However, this effect is weaker than the association between EAT volume and density. Therefore, an adjustment for EAT volume is necessary and in accord with other studies investigating the prognostic value of fat depots (e.g., Oikonomou et al. Lancet 2018). Our results show that higher EAT density at any given EAT volume is associated with mortality.

EAT volume and density are both independent predictors (see Model 1 in Table 3), and, therefore, both contribute to prognosis. From a CAD pathogenesis perspective, increased EAT volume has traditionally been associated with obesity and hyperlipidemia, while increased EAT density is a known index of inflammation, two separate pathways driving atherogenesis.

We added a corresponding description to the manuscript's results section and now list the univariable results in **Table 3**.

4. The result was concluded just from one dataset. Could the authors validate and replicate it using other cohorts like LIDC-IDRI and NSCLC Radiogenomics?

Thank you for raising the question of reproducibility and generalizability. Multiple other studies have described the prognostic value of EAT volume and density. Our study aimed to apply a biologically well-known concept in a specific cohort of patients eligible for lung cancer screening. Furthermore, the accuracy of the DL algorithm has been validated in an independent validation subset from PROMISE and NLST cohorts.

5. Is the conclusions just applied to groups of heavy smokers? Please discuss the situations in non-smoking and mild-smoking groups.

Thank you for bringing up this excellent point. This manuscript included participants from the NLST trial, all of whom have a ≥ 30 pack-year smoking history. This is an important population, as lung cancer screening is a common indication for noncontrast chest CT. Future work will be necessary to understand implications for non-smokers and lesser smokers – we've added the following sentence in italics to the Limitations section of the Discussion:

"Third, the NLST cohort included only heavy smokers (≥ 30 pack years), and differences in EAT's prognostic value could not be calculated between heavy smokers, non-smokers, and lower-risk smokers with ≥ 20 – 30 pack years, a group currently also recommended for lung cancer screening. *Future work is necessary to understand implications in non-smokers and lesser smoking populations.*"

Referee #2:

Please see attachment for a formatted reiew.

This manuscript shows that EAT volume and density from lung cancer screening images are associated with cardiovascular and all cause mortality. Effects are beyond clinical factors, BMI, and Agatston score. This is a significant contribution towards the use of chest CT images to obtain improved cardiovascular prediction. The manuscript needs considerable editing and additional analyses as it was difficult to follow.

Some specific concerns are:

- 1. There must be a better description of EAT segmentation. This is my biggest concern as all else flows from this. In the main document, it is not clear that this is the paper with "validation" of the method. An assertion was made with references 14 and 34. Neither one supports this claim. This led me on a wild goose chase trying to track down the validation. Then I found some validation in the Supplement. I had issues with the meager text.*

We thank the reviewer for their feedback and understand the concerns raised about the description of our EAT segmentation tool. In response, we have revised our manuscript to provide a more comprehensive and precise

description of our EAT segmentation process. We have moved the revised detailed explanation from the supplementary materials into the main body of the manuscript to ensure its visibility and accessibility.

Furthermore, we have made efforts to clarify that our paper serves as a validation of the EAT segmentation method. Specifically, we have emphasized our evaluation of the prognostic value of automatic EAT segmentation, which we believe is a crucial aspect of validating its utility and effectiveness. We hope these revisions address the reviewer's concerns and provide a clearer understanding of our work.

2. *I could not understand the text below. This system, used in the present study and in (4), is the third version of our heart segmentation system. The first (5) and second (6) versions were trained on 129 and 858 CT scans, respectively, and achieved median Dice coefficients of 0.90 and 0.95, respectively.*

We appreciate the reviewer's feedback on the clarity of our descriptions. In response, we have streamlined the information about the different versions of our heart segmentation system. Instead of detailing each version and their respective training data and performance, we have consolidated this information into a single sentence towards the end of the relevant section. This sentence briefly acknowledges the evolution of our system, focusing on the current version used in this study, which we believe provides a more straightforward and concise presentation of our work.

3. *What is the loss function?*

We employed the Dice loss function for training our models. We have incorporated this information into the updated description of our methodology.

4. *As this is the validation for the entire result, I would have expected some figures showing good segmentations and bad segmentations.*

Thank you very much for pointing out the missing image examples. Only a few (n=43) scans (see Flowchart **Supplemental Figure S1**) failed to be analyzed by the Deep Learning algorithm. We added image examples (**Supplemental Figure S2**) (see below) that display good and bad segmentations, including the most common reasons for failure. We reference the figure in the methods section of the manuscript.

	Axial	Coronal	Sagittal	3D - Heart	3D - EAT	Reason for failure
Optimal						N/A
Failed						- corrupted scan; incorrect spacing with overlapping images and image distortion
Failed						- corrupted scan; part of the heart is missing on the CT scan
Failed						- wrong field of view, CT scan ends at the level of the the ascending aorta and does not capture the heart
Failed						- incorrect image orientation (anterior vs. posterior)

5. *With the extensive down sampling, I am surprised that one can even see the pericardial sac. Is this a magic down-sampling technique?*

We appreciate the reviewer's question regarding our down-sampling technique. The extensive down-sampling was applied in the first step, a network tasked with localizing the heart through coarse segmentation. The scans are then cropped around the heart center to guarantee that the entire heart is captured.

The second network, which is responsible for the more detailed segmentation, operates on a suitably down-sampled spacing. This two-step process allows us to maintain the necessary detail for accurate segmentation while benefiting from down-sampling's computational efficiency.

6. *There is a lot of talk about versions and data. It is impenetrable. It probably makes perfect sense, but I do not understand it.*

We appreciate the reviewer's feedback regarding the complexity of our descriptions of the different versions and data. In response to this, as outlined in our answer to question 2, we have simplified this information in our revised manuscript to ensure it is more accessible and easier to understand.

7. *Was testing indeed done on a held-out test set of 1306 scans, all of which were volumetrically manually labeled?*

Yes. Overall, our group manually labeled 2,164 cases (PROMISE: n=1,140; FHS: n=628; NLST: n=396). Among these, 858 and 1,306 independent cases were used for training and testing, respectively. We adjusted the methods section of the manuscript to clarify this aspect.

8. *Was testing done at the original resolution or down-sampled resolution? If the original, a DICE of 0.95 hardly makes sense as the associated partial volume errors would probably give a worse DICE than this.*

We appreciate the reviewer's inquiry. Indeed, the testing was performed at the original resolution. The Dice score of 0.95, which we reported, is associated with heart segmentation. Such a score is achievable, especially considering the size of the dataset used for training. Our dice score is in a range of previously described dice scores for segmenting thoracic structures, as reported in the 2017 AAPM challenge (1). We revised the methods section to explicitly state that the evaluation was conducted on the original spacing/resolution.

1. <https://aapm.onlinelibrary.wiley.com/doi/10.1002/mp.13141>

9. *What is the loss function?*

Please refer to the answer to the question 3.

10. *It would be nice to compare EAT from lung cancer screening to a gated exam. Probably more is missing.*

We used the DL algorithm to segment EAT in the PROMISE cohort (Foldyna et al. JACC CVI 2021). PROMISE CT scans are ECG-gated. The mean EAT volume and density in PROMISE were $57.5 \pm 22.0 \text{ cm}^3/\text{m}^2$ and $-86.8 \pm 5.1 \text{ HU}$, respectively. The values are similar to those we measured in the NLST cohort ($70.3 \pm 24.6 \text{ cm}^3/\text{m}^2$ and $-77.7 \pm 5.2 \text{ HU}$), noting that the PROMISE cohort was substantially different. PROMISE enrolled symptomatic patients with stable chest pain, who were more frequently female (51% vs. 49%), and there were fewer patients with tobacco use (51%) compared to the 100% in NLST. PROMISE patients had a higher prevalence of traditional cardiovascular risk factors such as hypertension (64% vs. 35%) and diabetes mellitus (20% vs. 10%), limiting the comparison.

11. *As some scans will suffer from motion making it very difficult to identify the pericardial sac, I am surprised that authors did not simply lump EAT and paricardial adipose. For that matter, is there any association of risk with paricardial adipose?*

Pericardial (outside pericardial sac) and epicardial (inside pericardial sac) adipose tissue (EAT) are often described together as paracardial adipose tissue. Our algorithm is developed to recognize the pericardial sac and specifically measure the epicardial fat. EAT is known to have a stronger association with coronary artery disease and adverse events than other fat depots due to its stronger metabolic activity and proximity to the coronary arteries and myocardium. In our study, higher EAT volume was related to a higher CAC at baseline, as shown in **Table 2**. This association remained significant after a full adjustment for age, sex, race, ethnicity, smoking status (current vs. former), number of pack years, hx of myocardial infarction, hx of stroke, diabetes mellitus, hypertension, education status, and BMI (see full model results below). We added a corresponding description to the results section (Page 4).

ln_cac	Coefficient	Std. err.	t	P> t	[95% conf. interval]	
eat_ivol10_t0	.0222266	.0090193	2.46	0.014	.0045475	.0399057
age	.122872	.0041794	29.40	0.000	.1146798	.1310642
sex	1.279077	.0421313	30.36	0.000	1.196494	1.36166
race2						
2	-.5198149	.100549	-5.17	0.000	-.7169045	-.3227253
3	-.1955102	.1366133	-1.43	0.152	-.4632907	.0722703
4	-.182649	.1363321	-1.34	0.180	-.4498784	.0845804
ethnic2						
2	-.0088694	.1530421	-0.06	0.954	-.3088527	.291114
3	.5262974	.3553976	1.48	0.139	-.17033	1.222925
cigsmok	.1193501	.0412803	2.89	0.004	.0384352	.2002651
pkyr	.0046311	.0008754	5.29	0.000	.0029152	.0063469
diaghear	1.526601	.0616643	24.76	0.000	1.405731	1.647472
diagstro	.2366427	.1225957	1.93	0.054	-.0036615	.4769469
diagdiab	.467688	.0706894	6.62	0.000	.3291271	.6062488
diaghype	.4616502	.0440392	10.48	0.000	.3753274	.547973
edu						
2	-.1265718	.0646864	-1.96	0.050	-.2533659	.0002223
3	-.145421	.049144	-2.96	0.003	-.2417499	-.0490921
4	-.1063587	.0645384	-1.65	0.099	-.2328627	.0201453
5	-.1325295	.1549378	-0.86	0.392	-.4362285	.1711695
bmi	.0048255	.0044098	1.09	0.274	-.0038183	.0134694
_cons	-5.522332	.3326241	-16.60	0.000	-6.174321	-4.870344

12. Table 3. What is Model 3?

Thank you for pointing out the unclear description of **Table 3**. We adjusted the footnote to describe the individual Cox regression models correctly.

13. Table 4. How much of the incremental CAC value is due to the power of zero?

In the subgroup of NLST subjects with available CAC scores (N=13,966), 3,355 (24%) presented with CAC=0. We performed an additional Harrell's C concordance analysis, including a CAC yes/no variable instead of a categorical CAC variable in the nested models. The incremental value of CAC yes/no was slightly lower than a model including categorical CAC—Harrell's C: 0.703 vs. 0.706 for all-cause mortality and 0.757 vs. 0.769 for CV mortality.

14. Results. What is this? 10 cm³/m². At first, I thought it was normalized by height. But that does not make sense. Please explicitly describe.

Thank you for pointing out this unclear aspect of the methods section. The unit describes EAT volume (cm³) indexed by body surface area (m²). We describe the indexing method in the methods section. Per journal guidelines, the results section comes first. Therefore, we revised the results description to make the indexing method more transparent.

The size of fat depots is associated with the individual body habitus and obesity. In our cohort, there was a moderate correlation between the raw EAT volume (cm³) and BMI as well as BSA (BMI rho: +0.51, p<0.001; BSA rho: +0.56, p<0.001). Therefore, we chose an approach used in other studies that indexed the raw EAT volume by BSA and adjusted all multivariable models for BMI.

The corresponding description can be found in the Methods section on page 12.

15. *In some analyses, the EAT and HU HRs for cardiovascular death was higher than for all cause. Yet, when we look at Fig 3, there is a remarkable effect of EAT and HU on all cause! Are there any inconsistencies here? I would like to see some discussion on Fig 3 and all-cause death.*

Thank you for pointing out this unclear aspect. EAT volume and density are closely related to coronary artery disease and, therefore, are more prognostic for cardiovascular death. Our study found a positive independent association between EAT volume and coronary artery disease, as shown in response to comment #11.

The different magnitude of effect, comparing all-cause vs. CV mortality in Fig. 3, is due to the difference in event rates (All-cause mortality: 19.5% vs. CV mortality: 4.5%). Furthermore, EAT volume and density are categorized in quartiles in Fig. 3, while they are continuous in Table 3. Lastly, Fig. 3 is based only on univariable analysis, while Table 3 includes multivariable models. We checked the statistical analysis and did not find any inconsistencies.

Referee #3:

This research study investigated the prognostic value of deep learning-based automated epicardial adipose tissue (EAT) quantification in heavy smokers at risk for cardiovascular disease (CVD). The study compared the predictive capabilities of EAT quantification with traditional clinical risk factors and coronary artery calcium (CAC) scoring. The researchers analyzed data from 24,090 participants in the National Lung Screening Trial (NLST). These participants underwent non-ECG-synchronized, non-contrast low-dose chest computed tomography (CT) scans. A previously developed deep-learning algorithm (for ECG-gated CT) was employed to quantify the volume and density of epicardial adipose tissue (EAT).

Over a median follow-up period of 12.3 years, EAT volume and density were independently associated with all-cause and cardiovascular mortality. This suggests that assessing EAT volume and density through low-dose chest CT scans could be a useful tool for CVD risk stratification in heavy smokers, who are already at an increased risk for cardiovascular problems due to their smoking habits. The findings of this research study suggest that incorporating EAT quantification into routine lung cancer screening CT scans can provide valuable information for assessing the risk of cardiovascular disease and mortality in heavy smokers.

While the overall concept is worthwhile and the NLST cohort is appropriate for this study, there are several significant limitations, which need to be addressed, mostly related to the methods used and to the patient selection within the NLST cohort.

1. *A major and critical fault is that the EAT probability score is based on machine learning (even if it is a simple logistic regression) and is designed to maximize the mortality prediction, but it was developed in the same population it is subsequently tested in. This is likely causing a significant overfit. Importantly, this probability score is shown to be the key variable, but development and testing in the same cohort undermine its significance.*

Thank you for this comment and the opportunity to clarify our approach. First and foremost, our paper serves as a validation of the EAT segmentation method. As described in the methods section, a dataset of 2,164 randomly selected CT scans from FHS (n=628), PROMISE (n=1,140), and NLST (n=396) was used to develop the system. We used 858 cases (FHS, n=628, PROMISE, n=130; NLST, n=100) for training and tuning of the algorithm, while the remaining 1,306 cases (PROMISE, n=1,010; NLST, n=296) were reserved for testing.

We used a multivariable logistic regression model including EAT volume and density to calculate an EAT-based probability score, which we stratified into three risk categories (low: <15%, medium: 15%–25%, high: >25%). This

probability score is purely exploratory and should not be seen as a prediction score, which would need to fulfill specific requirements.

2. *Moreover, the marginal incremental AUC gain of 1% by adding EAT score for mortality prediction (is it significant by AUC ?) may be attributed from the leakage of the outcome data when fitting the EAT probability score.*

Please see our response to comment #1.

3. *The overall performance of clinical parameters alone is better than any of the imaging parameters. It is not clear that in Table 4 the clinical+CAC+EAT is better than clinical+CAC. Importantly EAT already includes information about mortality from the fitting as described in the appendix.*

Please see our response to comment #1.

4. *Why is the CAC score available on the subset only? Was the additional value of EAT compared on 13996/24090 images only? How was this subset selected? Previously the same group published a study in nature communications where 14,959 cases were processed for CAC scoring. How were the CAC scores obtained for this analysis? Seems many different sub-cohorts are utilized.*

Thank you for pointing out the CAC subgroup measurements. We utilized the validated CAC scores obtained by our group and published in Zeleznik et al. Nature Comm 2021. At that time, the policy of the National Cancer Institute was to only release up to 15,000 participants' data for any individual project.

In the current study, we applied stricter image quality check criteria than the Zeleznik et al. study. Consequently, we excluded a larger group of subjects before the EAT analysis (1,682/25,815 (6.5%)) vs. 41/15,000 (0.3%) by Zeleznik et al. Therefore, some subjects eligible for CAC scoring did not undergo EAT segmentation. We now provide a more comprehensive consort diagram to explain the individual steps to obtain the final analytical cohort of 24,090 cases.

5. *Were stents and pacemakers excluded from the CAC analysis?*

The analysis included all available NLST participants, not excluding those with any implants, to avoid selection bias.

6. *Since EAT volume and attenuation are negatively (and significantly) correlated, they shouldn't be in the same model. The authors do not present EAT attenuation in a model without EAT volume. EAT attenuation and volume should be presented separately.*

The correlation between EAT volume and density is a crucial aspect of the analysis and can be explained by the complex biology of the epicardial adipose tissue. Please refer to the response to Referee #1 Comment #3.

7. *There is no validation of quantification of EAT for nongated scans provided.*

We appreciate the reviewer's observation. We would like to clarify that our EAT segmentation system has indeed been developed and validated on ECG-gated and non-gated scans. Specifically, it was trained and tested, in addition to other cohorts, on an independent subset of the National Lung Screening Trial (NLST) cohort, which consists of non-gated low-dose CT scans. This information was previously detailed in the supplementary material and has now been incorporated into the main body of the revised manuscript for better visibility and understanding.

8. *The presented NRI is continuous and does not have a straightforward clinical interpretation. Can categorical positive/negative NRIs and 95% CI be provided?*

We agree with the reviewer that the continuous net reclassification index (NRI) is difficult to interpret. In general, NRI tests are increasingly criticized by the research community for their unacceptable statistical behavior, incorrect statistical inferences, and lack of interpretability (Kathleen F. Kerr Radiology 2022). Therefore, we would like to

abstain from showing 95% CIs. Furthermore, our regression models include multiple independent variables (categorical and continuous), so that we cannot provide categorical NRIs directly. Regarding the probability score, which we classified into three categories (low, medium, high), Kathleen F. Kerr et al. (Epidemiology, 2014) advise against using the NRI for 3 or more categories since it does not discriminate between different types of reclassifications.

Instead, we added Harrell's C statistics (AUC equivalent for time-to-event data) to test individual models' performance. Moreover, we revised **Table 4** to make it more readable and added Likelihood-ratio tests for nested models to test for incremental value of EAT and CAC vs. clinical parameters.

9. *Why is CAC modeled as a categorical variable, but EAT is modeled as continuous? Why?*

We adhered to the clinical standard of using CAC categories. However, a supplemental sensitivity analysis using CAC as a continuous measure did not reveal any differences in results.

Multivariable regression results were as follows:

All-cause mortality:

_t	Haz. ratio	Std. err.	z	P> z	[95% conf. interval]	
eat_ivol10_t0	1.103422	.0121176	8.96	0.000	1.079926	1.12743
eat_dens10_t0	1.376879	.0758867	5.80	0.000	1.235895	1.533945
age	1.083305	.0041897	20.69	0.000	1.075125	1.091548
sex	1.251919	.0540996	5.20	0.000	1.150252	1.362572
race2						
2	1.3622	.1225347	3.44	0.001	1.142016	1.624836
3	.6260857	.0924328	-3.17	0.002	.4687764	.8361839
4	1.208589	.152251	1.50	0.133	.9441683	1.547064
ethnic2						
2	1.443137	.2382599	2.22	0.026	1.04418	1.994526
3	1.581958	.4723113	1.54	0.124	.8811694	2.840081
cigsmok	1.775327	.0715067	14.25	0.000	1.640565	1.921158
pkyr	1.006447	.0006865	9.42	0.000	1.005102	1.007793
diaghear	1.315832	.0673892	5.36	0.000	1.190165	1.454769
diagstro	1.396459	.1226933	3.80	0.000	1.17555	1.65888
diagdiab	1.54958	.0868501	7.81	0.000	1.388374	1.729505
diaghype	1.121882	.0461297	2.80	0.005	1.035016	1.216037
edu						
2	.938259	.055468	-1.08	0.281	.8356057	1.053523
3	.7624207	.0349737	-5.91	0.000	.6968647	.8341438
4	.6814293	.0442918	-5.90	0.000	.5999211	.7740117
5	.8939959	.1285786	-0.78	0.436	.6743911	1.185112
bmi	.9860295	.0043711	-3.17	0.002	.9774994	.9946341
cac	1.000158	.0000198	7.98	0.000	1.000119	1.000197

CV-mortality:

_t	Haz. ratio	Std. err.	z	P> z	[95% conf. interval]	
eat_ivol10_t0	1.148502	.0248686	6.39	0.000	1.10078	1.198292
eat_dens10_t0	1.772464	.2020065	5.02	0.000	1.417642	2.216095
age	1.088121	.0089583	10.26	0.000	1.070704	1.105822
sex	1.280291	.1185036	2.67	0.008	1.067878	1.534956
race2						
2	1.824711	.2982972	3.68	0.000	1.324471	2.513885
3	.7217698	.2228484	-1.06	0.291	.3940832	1.321933
4	1.758804	.3959725	2.51	0.012	1.131307	2.734354
ethnic2						
2	1.579864	.5473468	1.32	0.187	.8011604	3.115445
3	3.675763	1.80614	2.65	0.008	1.403128	9.629372
cigsmok	2.022939	.1720444	8.28	0.000	1.712344	2.389873
pkyr	1.005285	.0014388	3.68	0.000	1.002469	1.008109
diaghear	1.86703	.1800163	6.48	0.000	1.545538	2.255396
diagstro	1.894935	.2877558	4.21	0.000	1.407133	2.551842
diagdiab	1.576242	.1703766	4.21	0.000	1.275311	1.948182
diaghype	1.288867	.1111598	2.94	0.003	1.088417	1.526234
edu						
2	.9955436	.1217683	-0.04	0.971	.783334	1.265242
3	.7889289	.0763435	-2.45	0.014	.6526319	.9536904
4	.6709273	.0939262	-2.85	0.004	.5099315	.8827528
5	.8342686	.2701527	-0.56	0.576	.4422522	1.573772
bmi	1.025107	.0086498	2.94	0.003	1.008293	1.042201
cac	1.000188	.0000334	5.64	0.000	1.000123	1.000254

10. Author affiliations missing for MCL, VKR, TM

Thank you for pointing out the missing information. We added the corresponding affiliations for MCL, VKR, and TM.

11. Table 3 - model 3 definition is missing (which covariates were adjusted?)

Thank you for pointing out the unclear description of Table 3. We adjusted the footnote of Table 3 to describe the individual Cox regression models clearly. Model 3 is the fully revised model, including EAT volume, EAT density, age, sex, Race, Ethnicity, smoking (current vs. former), pack-years, history of heart disease, stroke, diabetes mellitus, hypertension, education status, BMI, and CAC score.

12. Why is the distinction of CAC categories <300 HU (standard to have 400 HU as threshold)- misleading?

We used the standard density threshold of 130 HU to render CAC and the Agatston method to calculate the coronary calcium score. We calculated the CAC scores and stratified them into clinically relevant categories. While some studies use a threshold of 400 Agatston units, we used the 300 Agatston unit cutoff according to prior publications in primary prevention cohorts (e.g., Hoffmann et al. Am. J. Cardiol. 2008, Zeleznik et al. Nature Communications 2021). We added the corresponding reference (Hoffmann et al. Am. J. Cardiol. 2008) to the methods section.

13. For Table 4, what is the N value?

The N value is 13,860. We added a corresponding description to the **Table 4** footnote.

14. *In addition, the algorithm offers EAT measurements in under 2 seconds without human input, making it an "end-to-end" solution for CV risk assessment in clinical settings" -On what kind of a computer? Is this GPU enabled, and what kind of GPU? Does it include CAC scoring?*

We thank the reviewer for the question. The EAT segmentation tool does not include CAC scoring. It performs under 2 seconds on a GPU-enabled machine – specifically, on a Linux workstation with an Nvidia A6000 GPU. We added the missing information to the revised manuscript.

15. *Discussion - Inflammation is a common pathophysiology pathway for CVD and cancer -this paragraph is speculative; no data is presented in the manuscript relating to this. Same with the Epicardial adipose tissue section may be an index and promotor of local coronary artery inflammation. This paragraph is unrelated to the paper.*

Thank you for pointing out this unclear portion of the discussion. We agree that those two sections included information outside the scope of the current study. We condensed both paragraphs into one short section, added corresponding data from our study, and revised the references.

Reviewers' comments:

Reviewer #1 (Remarks to the Author):

The authors prepared appropriate answers to my questions and revisited their paper. Therefore, I think this manuscript has reached a level with no publication problems.

Reviewer #2 (Remarks to the Author):

Authors responded adequately to my concerns. This is a great example of opportunistic screening. Thank you for providing details on the segmentations.

David L Wilson

Reviewer #3 (Remarks to the Author):

Some progress has been made in the response, but unfortunately the authors did not answer the main critical points.

1. A major and critical fault is that the EAT probability score is based on machine learning (even if it is a simple logistic regression) and is designed to maximize the mortality prediction, but it was developed in the same population it is subsequently tested in. This is likely causing a significant overfit. Importantly, this probability score is shown to be the key variable, but development and testing in the same cohort undermine its significance.

>This probability is purely exploratory and should not be seen as a prediction score, which would need to fulfill specific requirements.

This EAT probability score features as the main result in the key figure 3 of the paper. Considering the guidelines for AI research and the high profile of this journal, this parameter should be developed and evaluated in separate cohorts or removed from the analysis. Consequently, points 2 and 3 and not answered at all as they merely refer to the answer 1 in point.

The answer to point 4 is unclear -if all 24090 cases could be processed for EAT, why could they not be processed for calcium? The paper is very confusing as Table 1 baseline characteristics have a different population than Table 4 . N is not given in Table 4. Also, CAC in Table 1 is incorrectly given as if for the whole population.

Regarding point 8 the authors still utilize continuous NRI and do not want to show categorical NRI based on an editorial from 2022. This editorial actually criticizes continuous NRI in particular. Regarding categorical NRI the editorial ignores the fact that one can show reclassification separately for events and nonevents, which should be standard practice. The categorical NRI is often used, and the original manuscript on categorical NRI is cited over 6270 times and over 653 times since 2022 according to Google Scholar.

Response to review

Report from reviewer 3

Some progress has been made in the response, but unfortunately the authors did not answer the main critical points.

1. A major and critical fault is that the EAT probability score is based on machine learning (even if it is a simple logistic regression) and is designed to maximize the mortality prediction, but it was developed in the same population it is subsequently tested in. This is likely causing a significant overfit. Importantly, this probability score is shown to be the key variable, but development and testing in the same cohort undermine its significance.

Since the authors used some 24k+ patients from a registry obtained from multiple sites, this should represent many different scanners and locations. This should help generalizability. Of course, a different sub-population might yield somewhat different results, due to calibration differences for example. I think it is quite OK as long as the authors describe this as a limitation.

More troubling is that I see no evidence of splitting their existing data into training/testing groups. Normally, we require this in Cox modeling. They do have a saving grace in that they are using very few features. However, when they include CAC + EAT volume + EAT HU + clinicals, they do have a number of features. It would be better practice to split out training and testing data and report performance on both. Maybe this was done. If so, it should be made more clear.

>This probability is purely exploratory and should not be seen as a prediction score, which would need to fulfill specific requirements.

This EAT probability score features as the main result in the key figure 3 of the paper. Considering the guidelines for AI research and the high profile of this journal, this parameter should be developed and evaluated in separate cohorts or removed from the analysis.

Consequently, points 2 and 3 and not answered at all as they merely refer to the answer 1 in point.

The answer to point 4 is unclear -if all 24090 cases could be processed for EAT, why could they not be processed for calcium? The paper is very confusing as Table 1 baseline characteristics have a different population than Table 4 . N is not given in Table 4. Also, CAC in Table 1 is incorrectly given as if for the whole population.

Regarding point 8 the authors still utilize continuous NRI and do not want to show

categorical NRI based on an editorial from 2022. This editorial actually criticizes continuous NRI in particular. Regarding categorical NRI the editorial ignores the fact that one can show reclassification separately for events and nonevents, which should be standard practice. The categorical NRI is often used, and the original manuscript on categorical NRI is cited over 6270 times and over 653 times since 2022 according to Google Scholar.

Authors could discuss categorical NRI and more carefully discuss the types of reclassifications.

This email has been sent through the Springer Nature Tracking System NY-610A-NPG&MTS

Point-by-point responses to the editor and reviewers

We thank the editor and the reviewers for their valuable feedback and insightful comments and appreciate the time and effort they have put into reviewing our manuscript. We also thank reviewer #2 for additional feedback. We agree with the suggestions and made the necessary modifications to the manuscript. Please find detailed responses to individual comments below:

Referee #1:

"The authors prepared appropriate answers to my questions and revisited their paper. Therefore, I think this manuscript has reached a level with no publication problems."

Thank you very much for the positive feedback.

Referee #2:

"Authors responded adequately to my concerns. This is a great example of opportunistic screening. Thank you for providing details on the segmentations."

We thank the reviewer for the positive feedback and are glad we could provide the requested information.

Referee #3:

"Some progress has been made in the response, but unfortunately the authors did not answer the main critical points."

A major and critical fault is that the EAT probability score is based on machine learning (even if it is a simple logistic regression) and is designed to maximize the mortality prediction, but it was developed in the same population it is subsequently tested in. This is likely causing a significant overfit. Importantly, this probability score is shown to be the key variable, but development and testing in the same cohort undermine its significance.

This probability is purely exploratory and should not be seen as a prediction score, which would need to fulfill specific requirements."

We thank the reviewer for pointing out this unclear aspect and apologize for any confusion.

Epicardial adipose tissue (EAT) volume and density are known prognostic markers, tested and validated in other large imaging studies, as mentioned by Mancio et al. EHJ CVI 2018. Our discussion states the following:

"Our study corroborates growing evidence linking increased EAT volume and density with adverse events. For instance, a large meta-analysis including over 20,000 subjects (mainly from the Framingham Heart Study (FHS), Multi-Ethnic Study of Atherosclerosis (MESA), Heinz Nixdorf Recall (HNR) study, EISNER study, and the Rotterdam study) reported a strong relationship between EAT volume, CV risk factors, and CAD severity²¹."

In the current study, we automated the EAT segmentation. We validated EAT's prognostic value as an opportunistic imaging marker in a cohort of individuals who underwent lung cancer screening CTs, a large cohort where manual segmentation is not feasible. The machine learning was used solely for the segmentation of the EAT. It was trained and validated in the Framingham Heart Study, PROMISE, and NLST, a variety of large, well-phenotyped cohorts with images acquired using different scanners at numerous sites across the US. Hence, our study did not intend to develop and test a novel biomarker; instead, it used deep learning to assess a well-known and validated imaging marker in a cohort of patients with increased CV risk who usually do not get CV evaluation.

The methods section states the following:

Training and testing data sets

A dataset of 2,164 randomly selected CT scans from FHS (n=628), PROMISE (n=1,140), and NLST (n=396) was used to develop the system. Four experienced cardiovascular radiologists provided standard manual segmentations for all 2,164 cases.

We used 858 cases (FHS, n=628, PROMISE, n=130; NLST, n=100) for training and tuning of the algorithm, while the remaining 1,306 cases (PROMISE, n=1,010; NLST, n=296) were reserved for testing.

“This EAT probability score features as the main result in the key figure 3 of the paper. Considering the guidelines for AI research and the high profile of this journal, this parameter should be developed and evaluated in separate cohorts or removed from the analysis.

Consequently, points 2 and 3 and not answered at all as they merely refer to the answer 1 in point.”

Again, we apologize for the confusion. As mentioned above, we did not define a new biomarker. We agree that the exploratory “EAT probability score” may need further validation in independent cohorts. We added the following sentence to the limitation section of the discussion.

“Fourth, the EAT probability score combining EAT volume and density needs external validation. However, the large number of subjects from multiple sites (i.e., various scanners at different locations) warrants the generalizability of our results.”

“The answer to point 4 is unclear -if all 24090 cases could be processed for EAT, why could they not be processed for calcium? The paper is very confusing as Table 1 baseline characteristics have a different population than Table 4 . N is not given in Table 4. Also, CAC in Table 1 is incorrectly given as if for the whole population.”

Thank you for pointing out the CAC measurements available in a subgroup of the NLST participants. CAC scores were used solely for the multivariable adjustment in our study. We utilized the validated CAC scores obtained by our group and published in Zeleznik et al. Nature Comm 2021. As mentioned in our previous rebuttal, at the time of the last study, the policy of the National Cancer Institute was to only release up to 15,000 participants' data for any individual project. We did not re-segment the cohort as a newer CAC segmentation algorithm is being developed, and we decided to use the validated results derived from analyses performed in a strictly controlled fashion. The CAC scores in the 13,996 individuals employed in the current study represent a random sample (58%) of the NLST participants.

Moreover, since all major results were statistically significant in this random sample, we expect that a larger sample size would lead to a lower standard error and, therefore, would further strengthen our findings by making the current results even more significant. However, we added a corresponding disclosure to the limitations section of the manuscript, stating that CAC was available solely in a random subgroup of the NLST cohort. This information is also available in the methods section of the manuscript and in the footnote of Tables 1 & 2 to avoid confusion. Furthermore, we added corresponding sample sizes for individual models to the footnote of Table 4, as requested by the reviewer.

“Regarding point 8 the authors still utilize continuous NRI and do not want to show categorical NRI based on an editorial from 2022. This editorial actually criticizes continuous NRI in particular. Regarding categorical NRI the editorial ignores the fact that one can show reclassification separately for events and nonevents, which should be standard practice. The categorical NRI is often used, and the original manuscript on categorical NRI is cited over 6270 times and over 653 times since 2022 according to Google Scholar.”

We thank the reviewer for the comment and the opportunity to discuss our approach in more detail. The reviewer is correct in stating that the beforementioned editorial by Kathleen F. Kerr (Radiology, 2022) criticizes the category-free (i.e., continuous) NRI. The editorial states, “the category-free NRI statistic ignores the magnitude of change”. However, the editorial also states that the “same issue can arise with categorical NRI statistics because they also only account for the direction of risk reclassification, not the magnitude”. Moreover, we allow us to disagree with the reviewer's statement that the editorial ignores the separate reclassification for events and nonevents. Figure 1 and examples 1 to 3 differentiate between event and non-event NRIs.

The reviewer further states that “*categorical NRI is often used, and the original manuscript on categorical NRI is cited over 6270 times and over 653 times since 2022 according to Google Scholar*”. This statement is correct. However, the same author group that developed the highly cited original NRI (i.e., Pencina et al.) published the extensions of the NRI, i.e., the category-free or continuous NRI, in Statistics in Medicine in 2011. They state that the NRI “is more objective and comparable across studies if using the category-free version” and conclude that “the category-less or continuous NRI is the most objective and versatile measure of improvement in risk prediction”. They argue that “in cases where no established categories exist, it is more prudent to use a version of NRI which does not require categories, rather than trying to create them for one particular application”. This reflects our case.

Moreover, our analysis is of an exploratory nature, and future studies are needed to validate the results in different cohorts. As Pencina et al. (2011) point out: “the category-based NRI is influenced by the relationship between category cut-offs and event rates. Hence, it may be misleading to apply the same fixed categories to events defined differently or time horizons of different duration which lead to varying incidence rates. This problem is absent when we use the category-free NRI, which is unaffected by event rates”.

In summary, Pencina et al., who developed both the original and the continuous NRI, conclude that “the continuous NRI offers the widest and most standardized application.”

We added the following statement to the statistical analysis portion of the methods section:

“We used the continuous NRI as recommended by Pencina et al., as continuous NRI offers the widest and most standardized application and is not affected by different event rates and should thus be used when comparing NRIs across studies.”

Reviewers' comments:

Reviewer #3 (Remarks to the Author):

I had the opportunity to review the revised version of the manuscript. Unfortunately, the authors have not addressed some of the remaining issues.

In particular - "Importantly, this probability score is shown to be the key variable, but development and testing in the same cohort undermine its significance."

The most pressing concern is the methodology used for the probability score that combines density and volume. Authors' explanation does not acknowledge this and only discusses training for deriving EAT volumes. The concern is about the probability score, not the volumes.

The authors put the description of the probability score in the statistical section, but deriving the probability score is a form of machine learning even if done with multivariable logistic regression, and training data needs to be separated from the test data.

As previously highlighted, the approach of deriving and testing this combined score within the same population is methodologically flawed and has not been sufficiently addressed in the revision.

Therefore, the probability score should be removed from the manuscript or retrained using a separate population to eliminate any potential data leakage to the testing data.

The current approach, where the logistic regression model for the probability score is derived, trained, and tested within the same cohort, is not correct and likely overfits the data.

Regarding different sample sizes for CAC and EAT I understand the authors' explanation of different sample sizes for CAC and EAT due to the historical processing sequence, but it makes the current manuscript difficult to follow.

Point-by-point responses to the editor and reviewers

We thank the editor and the reviewers for their valuable feedback and insightful comments and appreciate the time and effort they have put into reviewing our manuscript. We also thank reviewer #3 for additional feedback. We agree with the suggestions and made the necessary modifications to the manuscript. Please find detailed responses to the comments below:

Reviewer #3 (Remarks to the Author):

"I had the opportunity to review the revised version of the manuscript. Unfortunately, the authors have not addressed some of the remaining issues. In particular - "Importantly, this probability score is shown to be the key variable, but development and testing in the same cohort undermine its significance."

The most pressing concern is the methodology used for the probability score that combines density and volume. Authors' explanation does not acknowledge this and only discusses training for deriving EAT volumes. The concern is about the probability score, not the volumes.

The authors put the description of the probability score in the statistical section, but deriving the probability score is a form of machine learning even if done with multivariable logistic regression, and training data needs to be separated from the test data.

As previously highlighted, the approach of deriving and testing this combined score within the same population is methodologically flawed and has not been sufficiently addressed in the revision.

Therefore, the probability score should be removed from the manuscript or retrained using a separate population to eliminate any potential data leakage to the testing data.

The current approach, where the logistic regression model for the probability score is derived, trained, and tested within the same cohort, is not correct and likely overfits the data.

Regarding different sample sizes for CAC and EAT I understand the authors' explanation of different sample sizes for CAC and EAT due to the historical processing sequence, but it makes the current manuscript difficult to follow."

Response:

We thank the reviewer for pointing out their concerns regarding developing and testing the probability score.

As proposed by the reviewer, we removed the score from the manuscript and now provide data only on the EAT volume and density as separate variables. We now offer additional adjusted KM curves and recalculated the statistics in the revised Table 4 using EAT volume and density as individual variables. We also removed the description of the probability score from the results, methods, and the corresponding Supplemental Text S1, score-related KM curves (Figure 3), and Supplemental Tables S4 & S5. Finally, we also adjusted the discussion accordingly.

REVIEWERS' COMMENTS:

Reviewer #3 (Remarks to the Author):

authors finally resolved the issue